

# Stress coping strategies used by nurses during the COVID-19 pandemic

Matylda Sierakowska[1] and Halina Doroszkiewicz[2]

[1] Department of Integrated Medical Care, Faculty of Health Sciences, Medical University of Bialystok, Bialystok, Poland

[2] Department of Geriatrics, Faculty of Health Sciences, Medical University of Bialystok, Białystok, Poland

Corresponding author
Matylda Sierakowska,
matylda.sierakowska@umb.edu.pl

## ABSTRACT

**Introduction**. The risk of getting SARS-CoV-2 infection, worries about exposing loved ones, anxiety and frustration, emotional and physical exhaustion, burn out, a feeling of being overwhelmed, and struggles and challenges with parenting are a few among many factors that affect nurses' personal lives and professional functioning. The aim of the research is to assess nurses' level of stress during the COVID-19 pandemic and their sense of self-efficacy, and to learn what coping strategies they use.

**Methodology/Methods**. The study was carried out online and based on the diagnostic poll method, using an original survey questionnaire, the General Self-Efficacy Scale (GSES 10–40), the Mini-COPE questionnaire (0–3), and the Perceived Stress Scale PSS-10 (0–40). The respondent group was made up of nurses ($n = 220$) who provide health services in inpatient and outpatient health care institutions in the northeastern region of Poland. The statistical analysis was performed using the STATISTICA 13.0 package (StatSoft). The distribution of variables was checked with Shapiro-Wilk tests. The Mann–Whitney U test was used to compare two independent samples, while the Kruskal–Wallis test was used to compare more samples. The adopted statistical significance level was $p < 0.05$. Multivariate regression analysis was applied to determine which factors were related to the level of stress.

**Results**. The mean age of the participants was 43.3 years. The vast majority were women (96.4). The mean work experience of the nurses was over 20 years (58.2%). A total of 62.3% worked directly with patients in hospitals, including 11.8% respondents working at COVID-19 units and 37.7% working at primary care institutions. The analyses show that the respondents represented a high level of stress (PSS-10 – 20.9), related to their work experience as a nurse (β −0.250, $p = 0.014$), the number of hours worked a month (β 0.156, $p = 0.015$), and self-assessed health status (β −0.145, $p = 0.037$). They declared an average sense of self-efficacy (GSES – 29.1), which significantly depended on the nurses' places of employment ($p = 0.044$). Out of stress coping strategies (Mini-COPE), the younger nurses mentioned venting ($p = 0.010$), instrumental support ($p = 0.011$), sense of humour ($p = 0.013$) and self-blame (0.031). Practice nurses also chose the strategy of behavioral disengagement ($p = 0.032$), and nurse managers chose the strategy of planning ($p = 0.018$).

**Conclusions**. The experience of the COVID-19 pandemic highlights the need to implement some strategies to protect nurses' mental health and to take extensive prevention measures in critical situations. Special attention should be given to nurses who are younger and have shorter work experience. It is also important to monitor nurses' working time and health status, and those who work at outpatient health care institutions should be given more support and information.

## INTRODUCTION

The COVID-19 pandemic affects the mental health and social behaviors of people all over the world. Quick transmission of the virus resulting from globalization and people's mobility has caused global traumatic stress (*Heitzman, 2020*). The sense of instability and uncertainty connected with the pandemic, the need to change our plans, potential job loss and financial insecurity combined with social isolation have generated the feeling of losing control of our lives (*Ajduković et al., 2021*; *Giorgi et al., 2020*).

Apart from elderly people, children, adolescents and persons with mental disorders, the group that reacted very strongly to the COVID-19 situation was health care professionals (*Kennedy, 2021*). Working in pandemic conditions has been a special challenge for physicians, nurses, and other medical professionals, even though in many regions the pandemic has largely subsided and restrictions have been lifted. By doing their everyday tasks, medical professionals run the risk of losing their lives or health, which results in intense daily stress (*Pang et al., 2021*; *Giorgi et al., 2020*; *Ali et al., 2020*). The psychological impact of COVID-19 also includes burnout affecting health care workers (*Doolittle, Anderssen & Perreaux, 2020*).

It has been observed that nurses have some dilemmas as to whether they are willing to continue doing their job as a result of fatigue and fear of potential infection and its medical consequences. This is highlighted in the International Council of Nurses (ICN) report, showing that nurses all over the world are currently experiencing a psychological trauma, which may ultimately cause a direct threat to the nursing profession and health care systems (*Doolittle, Anderssen & Perreaux, 2020*). The ICN admits that stress experienced during the COVID-19 pandemic affects more than 50% American nurses; 49% Brazilian nurses report anxiety and 25%, depression; 60% Chinese nurses feel exhausted, and 90% suffer from anxiety; Spanish nurses report symptoms of anxiety and growing burnout; and 40% Israeli nurses are afraid of providing care for COVID-19 patients. The International Council of Nurses anticipates a global deficit of nursing staff in the upcoming decades, and the traumatic effect of the COVID-19 pandemic may significantly contribute to this phenomenon (*Doolittle, Anderssen & Perreaux, 2020*).

In the recent decades, instead of focusing on the experience of stress itself, researchers have shifted their focus people's ways of coping with stressful events, *i.e.,* coping strategies (*Lazarus & Folkman, 1984*). *Endler & Parker (1990)* identify three styles of coping: problem-focused (making efforts to solve the problem), emotion-focused (concentrating on oneself and one's own emotional experiences), and avoidance-oriented (avoiding the problem by engaging in substitute tasks or seeking social contacts).

It has been found that flexibility and modifying the applied strategies after evaluating their effectiveness are important elements of coping. According to Schwarzer, the author of the original version of the General Self-Efficacy Scale (GSES), effective coping in the face

of problems and challenges occurring in one's environment, as well as self confidence, are related to the sense of self-efficacy (*Schwarzer & Jerusalem, 1995*).

A low sense of self-efficacy is believed to result in anxiety and helplessness, while a high one helps people accept challenges, set goals, and succeed in achieving them (*Simonds, 2003*; *McCabe, Gilmartin & Goldsamt, 2016*). It is an important factor supporting the person's self-control capacity (*Bray, Graham & Saville, 2015*; *Najmi et al., 2013*), and according to *Kwak & Hur (2019)*, it ensures quick, proper, safe and accurate nursing interventions in emergencies and caring for patients in critical condition. The sense of self-efficacy may also be a factor protecting individuals from the negative consequences of stress (*Ogińska-Bulik & Juczyński, 2010*).

In the face of the COVID-19 pandemic and the risk of SARS-CoV-2 infection, many elements appear to have a negative impact on nurses' professional functioning. From the point of view of the managing staff, it is important to recognize these factors and effectively counteract them and plan future strategies of crisis management.

## THE AIM OF THE STUDY

The aim of the research is to assess nurses' level of stress during the COVID-19 pandemic, to identify stress-related factors, to assess the nurses' sense of self-efficacy, and to find out their coping strategies.

The authors are looking for answers to the following specific questions: what socio-demographic, personal and professional factors may be associated with nurses' psychological coping during the COVID-19 pandemic and what support and organizational activities may be taken to ensure better and safer working conditions.

The authors' research hypothesis is that work during the COVID-19 pandemic contributes to the deterioration of nurses' mental health.

## MATERIAL AND METHODS

### Participants and procedures

The study was carried out among Polish nurses who provide health services in inpatient and outpatient health care institutions, mainly in the northeastern part of Poland (Podlaskie Province). The total study group included 220 nurses.

The cross-sectional study was carried out between December 1, 2020, and February 28, 2021, among nurses working at hospitals and at primary health care facilities in the northeastern region of Poland. Invitations to participate in the study were sent to e-mail addresses of 400 nurses, and 220 respondents returned the questionnaires, so the response rate was 55%. We do not know the reasons for the other180 nurses choosing not to participate.

The study was carried out using an on-line questionnaire prepared in Google Forms in the Polish language. The participants' responses were recorded on the used Google platform, and then, the raw data in Excel were collected for statistical analysis. The time needed to complete the questionnaire was approx. 20 minutes.

Not currently working as a nurse (as a result of loss of the licence or a temporary break in professional activity) was the exclusion criterion. Each respondent could withdraw from the study at any point. Participation was entirely voluntary and anonymous, and no incentives were used.

## Ethics approval

The research was carried out in accordance with the Declaration of Helsinki and the Good Clinical Research Practice. The Bioethics Committee of the Medical University in Bialystok, Poland, granted the ethical approval for the study (APK.002.150.2020, APK.002.151.2021). Participation in the study was voluntary, and all the participants were informed about the research project and expressed their written consent to participate.

## Study instruments

The research was carried out with the method of diagnostic poll and using the following research tools:

- a self-designed survey questionnaire (17 questions), including demographics, data referring to the workplace, work experience, safe working conditions (Likert scale from 1 to 5), atmosphere in the medical team, opportunities to receive emotional and information support, and subjective health status assessment (very good, good, average, very poor, poor);
- the General Self-Efficacy Scale (GSES) by *Schwarzer & Jerusalem (1995)*, Polish adaptation: *Juczyński (2012)*;
- the Mini-COPE Inventory by Ch. Carver, Polish adaptation: *Juczyński & Ogińska-Bulik (2012a)*;
- the Perceived Stress Scale PSS-10 by S. Cohen, T. Kamarck and R. Marmelstein, Polish adaptation: *Juczyński & Ogińska-Bulik (2012b)*.

**The General Self-Efficacy Scale (GSES)** is a research tool including 10 questions used to test an individual's general belief in their efficacy in coping with difficulties and obstacles. The sum of all the obtained points gives a general index between 10 and 40 (the higher the score, the higher the sense of self-efficacy). The general score is then converted into standardized units and interpreted. The sten scores of 1–4 are treated as low, 5–6 as average, and 7–10 as high (*Juczyński, 2012*).

**The Mini-COPE Inventory** is used to measure dispositional coping, *i.e.,* to assess typical ways of experiencing and reacting to intense stress. Mini-COPE includes 28 items making 14 scales, which correspond to stress coping strategies: active coping, planning, positive reframing, acceptance, humor, religion, use of emotional support, use of instrumental support, self-distraction, denial, venting, substance use, behavioral disengagement, self-blame. The higher the score, the higher the intensity of applying the particular strategy (*Juczyński & Ogińska-Bulik, 2012a*).

**The Perceived Stress Scale PSS-10** includes 10 questions concerning events connected with stress, personal problems and feelings related to various life situations. The total score of the scale (after the conversion of positive questions) is the sum of all the obtained points (from 0 to 40 points). The higher the score, the greater the stress.

The general score is then converted into standardized units and interpreted. The sten scores of 1–4 are treated as low, 5–6 as average, and 7–10 as high (*Juczyński & Ogińska-Bulik, 2012b*).

## DATA ANALYSIS

The statistical analysis was performed using the STATISTICA 13.0 package (StatSoft). The distribution of variables was checked with Shapiro-Wilk tests. They were presented as frequency and percentage (categorical variables), as means and standard deviation, medians, and interquartile range (continuous variables). The Mann–Whitney U test was used to compare two independent samples, and the Kruskal–Wallis test was used for a higher number of samples. The adopted statistical significance level was $p < 0.05$.

Multi-factor regression analysis was applied to determine the factors correlated to the level of stress. Eleven variables were included in the model: age, work experience, education level, number of hours worked a month, assessment of atmosphere at work, sense of security, self-assessment of health status, place of care provision, nursing role, GSES index, and being informed by the employer about the current guidelines/recommendations regarding the SARS-CoV-2 threat. The analysis presented the estimations of parameters (β) with confidence intervals (95% CI) and p values. For the factors statistically significantly correlated with the level of stress, the *p* value is p<0.05.

## RESULTS

### Socio-demographic and occupational characteristics of the studied group of nurses

The mean age of the participants was 43.3 years (±12.6). The vast majority were women (96.4 %), and comparable proportions of the respondents had bachelor's and master's degrees (>40%). The mean work experience of the nurses was over 20 years (±13.7), and 80.5% worked directly with patients (as practice nurses). Nurse managers accounted for less than 20% of the group. More than one-third (37.7%) were nurses working at primary or specialist outpatient health care facilities. 11.8% respondents were working at COVID-19 units.

The vast majority (>70%) of the respondents assessed their own health as good or very good. On average, the nurses worked 178 hours a month, compared to the average of 168 in 2020 (standard: 37 h 55 min./week).

They assessed the security at the workplace as slightly above the mean (3.5 in the 1 to 5 scale). According to over 70% respondents, their employers informed them about the current guidelines concerning the SARS-CoV-2 and provided personal protection equipment to ensure safe performance of their job responsibilities (data in Table 1).

### General assessment of the level of stress, sense of self-efficacy and strategies of coping with stress

Data obtained in the analysis shows that the nurses (n = 220) had a high level of stress (PSS-10 –20.9). In sten interpretation, this is a sten score of 7. They also declared average general self-efficacy (GSES –29.1)—a sten score of 6.

**Table 1  Socio-demographic and occupational characteristics of the studied nurses ($n = 220$).**

| Variables studied | | Min. | Max. | Me | (Q1-Q3) |
|---|---|---|---|---|---|
| **Age in years**, $\bar{x} \pm$ SD | $43.3 \pm 12.6$ | 21 | 66 | 47 | 31–52 |
| **Gender**, n (%) women | 212 (96.4) | | | | |
| **Marital status**, n (%) married | 172 (78.2) | | | | |
| **Education**, n (%) | | | | | |
| master's degree | 101 (45.9) | | | | |
| bachelor's degree | 92 (41.8) | | | | |
| secondary | 27 (12.3) | | | | |
| **Working area**, n (%) | | | | | |
| Primary health care | 83 (37.7) | | | | |
| Intensive Care Unit | 21 (9.5) | | | | |
| COVID unit | 26 (11.8) | | | | |
| Non-invasive treatment unit | 43 (19.5) | | | | |
| Surgery unit | 47 (21.5) | | | | |
| **Work experience in years**, $x \pm$ SD | $20.9 \pm 13.7$ | 0.5 | 45 | 25 | 6–32 |
| 1–5 n (%) | 52 (23.6) | | | | |
| 6–20 n (%) | 40 (18.2) | | | | |
| >20 n (%) | 128 (58.2) | | | | |
| **Average number of working hours/month**, $x \pm$ SD | $179 \pm 48.1$ | 15 | 360 | 169 | 160–200 |
| **Health self-assessment**, n (%) | | | | | |
| very good | 156 (70.9) | | | | |
| average | 53 (24.1) | | | | |
| very poor/poor | 11 (5.0) | | | | |
| **Assessment of security at work (1–5)**, mean $\pm$ SD | $3.5 \pm 1.0$ | | | | |
| **Does the employer provide information on the current guidelines/recommendations concerning SARS-CoV-2?** Yes, n (%) | 159 (72.3) | | | | |
| **Does the employer provide PPE for safe performance of job responsibilities?** Yes, n (%) | 173 (78.6) | | | | |

**Notes.**

$x$, mean; $\pm$ SD, –standard deviation; Min., minimum; Max., maximum; M, median; Q1, lower quartile; Q3, upper quartile.

Generally, the most popular strategies of coping with stress (Mini-COPE) were active coping ($2.2 \pm 0.5$) and planning ($2.2 \pm 0.6$), as well as emotional support ($2.1 \pm 0.6$). The lowest score was obtained for substance use ($0.3 \pm 0.6$) (data in Table 2).

**Table 2 Mean scores in the PSS-10, GSES, and Mini-COPE ($n = 220$).**

| Variables studied | $x \pm SD$ |
|---|---|
| PSS-10 (0–40) | $20.9 \pm 5.2$ |
| GSES (10–40) | $29.1 \pm 3.6$ |
| Mini-COPE–strategies of coping with stress (0–3): | |
| Active coping | $2.2 \pm 0.5$ |
| Planning | $2.2 \pm 0.6$ |
| Positive reframing | $1.8 \pm 0.7$ |
| Acceptance | $1.9 \pm 0.6$ |
| Humor | $0.8 \pm 0.6$ |
| Religion | $1.5 \pm 0.9$ |
| Emotional support | $2.1 \pm 0.6$ |
| Instrumental support | $1.8 \pm 0.7$ |
| Self-distraction | $1.7 \pm 0.6$ |
| Denial | $0.8 \pm 0.6$ |
| Venting | $1.4 \pm 0.6$ |
| Substance use | $0.3 \pm 0.6$ |
| Behavioral disengagement | $0.8 \pm 0.6$ |
| Self-blame | $1.4 \pm 0.8$ |

**Notes.**
$x$, mean; $\pm$ SD, standard deviation; PSS-10, Perceived Stress Scale; GSES, Generalized Self-Efficacy Scale; Mini-COPE, Inventory for Measuring Coping With Stress.

## The level of stress, sense of self-efficacy, and coping strategies with respect to age, work experience, nursing role and place of care provision

The results show that the level of stress (PSS-10) of the studied nurses was not related to their age, work experience, nursing role, or the place of care provision (data in Tables 3–6).

Out of the above-mentioned variables, the sense of self-efficacy (GSES) was only significantly related to the respondents' workplace ($p = 0.044$). Nurses working at hospital declared a higher sense of self-efficacy than did nurses working at outpatient health care facilities (data in Table 6).

The study results concerning correlations between the respondents' age and the strategies of coping with stress (Mini-COPE) (data in Table 3) show a statistically significant relationship between the respondents' age and the strategy of venting ($p = 0.010$), instrumental support ($p = 0.011$), humor ($p = 0.013$), and self-blame ($p = 0.031$). These strategies were significantly more often chosen by younger nurses (under 40 years old).

Similar strategies were also correlated to the respondents' work experience (data in Table 4). Nurses with work experience up to 20 years declared coping strategies significantly correlated with the strategy of instrumental support ($p = 0.005$), humor ($p = 0.013$), and venting ($p = 0.010$). The strategy of self-blame ($p = 0.031$) was used by nurses with work experience up to 5 years.

The analysis of correlations between the nurses' professional role and coping strategies showed a statistically significant relationship with the strategy of planning ($p = 0.018$), behavioral disengagement ($p = 0.032$), and venting (p $= 0.033$). Nurse managers

**Table 3  The level of stress, coping strategies and sense of self-efficacy by the respondents' age ($n = 220$).**

| Variables studied | Age in years | | | |
| --- | --- | --- | --- | --- |
| | ≤40 ($n = 75$) | 41–50 ($n = 70$) | ≥50 ($n = 75$) | *p*-value |
| PSS-10, mean ± SD (0–40) | 24.5 ± 3.4 | 25.2 ± 3.8 | 24.7 ± 3.8 | 0.717 |
| GSES, mean ± SD (10–40) | 29.0 ± 3.6 | 28.8 ± 3.3 | 29.5 ± 3.8 | 0.903 |
| Mini-COPE – strategies of coping with stress (0–3): | | | | |
| Active coping | 2.2 ± 0.6 | 2.2 ± 0.5 | 2.3 ± 0.5 | 0.739 |
| Planning | 2.1 ± 0.7 | 2.1 ± 0.6 | 2.3 ± 0.5 | 0.166 |
| Positive reframing | 1.7 ± 0.7 | 1.8 ± 0.6 | 1.9 ± 0.7 | 0.469 |
| Acceptance | 1.7 ± 0.5 | 1.6 ± 0.5 | 1.6 ± 0.6 | 0.217 |
| Humor | 1.0 ± 0.7 | 0.8 ± 0.5 | 0.7 ± 0.5 | 0.013[1] |
| Religion | 1.4 ± 1.0 | 1.5 ± 1.1 | 1.5 ± 0.8 | 0.681 |
| Emotional support | 2.1 ± 0.6 | 2.0 ± 0.5 | 2.1 ± 0.5 | 0.228 |
| Instrumental support | 2.0 ± 0.8 | 1.8 ± 0.7 | 1.7 ± 0.6 | 0.011[2] |
| Self-distraction | 1.7 ± 0.6 | 1.7 ± 0.6 | 1.8 ± 0.6 | 0.799 |
| Denial | 0.8 ± 0.5 | 0.9 ± 0.6 | 0.8 ± 0.7 | 0.692 |
| Venting | 1.6 ± 0.6 | 1.4 ± 0.6 | 1.3 ± 0.6 | 0.010[3] |
| Substance use | 0.4 ± 0.7 | 0.3 ± 0.5 | 0.4 ± 0.6 | 0.736 |
| Behavioral disengagement | 0.9 ± 0.7 | 0.9 ± 0.7 | 0.8 ± 0.6 | 0.720 |
| Self-blame | 1.6 ± 0.9 | 1.3 ± 0.7 | 1.3 ± 0.7 | 0.031[4] |

Notes.

The *p*-value shows the significance of particular characteristics in groups of nurses.

In the case of compared groups, the Kruskal–Wallis test was used.

PSS-10, Perceived Stress Scale; GSES, Generalized Self-Efficacy Scale; Mini-COPE, Inventory for Measuring Coping With Stress.

Corrected the *p*-values for the multiple comparisons:

1) The differences refer to groups $< 40$ and $\leq 50$ ($p = 0.013$)

2) The differences refers to groups $< 40$ and $\leq 50$ ($p = 0.021$)

3) The differences refers to groups $< 40$ and $\leq 50$ ($p = 0.012$)

4) The differences refers to groups $< 40$ and $40$–$50$ ($p = 0.095$) and $< 40$ and $\leq 50$ ($p = 0,068$)

significantly more often chose the strategy of planning, and practice nurses more often chose behavioral disengagement and venting (data in Table 5).

With regard to the place of care provision (data in Table 6), it was found that clinical care nurses significantly more often chose the strategy of planning than did primary health care nurses ($p = 0.024$). It was also shown that primary health care nurses more often used the strategy of behavioral disengagement ($p = 0.041$).

In the next stage of analyses, the researchers used multivariate regression analysis to identify the factors connected with a higher level of stress among the studied nurses. Eleven variables were included in the model (age, work experience, education level, number of hours worked a month, assessment of atmosphere at work, sense of security, self-assessment of health status, place of care provision, nursing role, GSES index, and being informed by the employer about the current guidelines/recommendations regarding the SARS-CoV-2). The final model revealed three main predictors: work experience ($\beta$ −0.250, $p = 0.014$), the number of hours worked a month ($\beta$ 0.156, $p = 0.015$), and self-assessment of health

**Table 4  The level of stress, coping strategies and sense of self-efficacy by the respondents' professional role ($n = 220$).**

| Variables studied | Work experience in years | | | |
|---|---|---|---|---|
| | 1–5 ($n = 52$) | 6–20 ($n = 40$) | ≥20 ($n = 128$) | *p*-value |
| PSS-10, mean ± SD (0–40) | 24.9 ± 2.9 | 24.4 ± 4.2 | 24.8 ± 3.4 | 0.647 |
| GSES, mean ± SD (10–40) | 28.9 ± 3.9 | 29.3 ± 2.6 | 29.2 ± 3.7 | 0.691 |
| Mini-COPE – strategies of coping with stress (0–3): | | | | |
| Active coping | 2.2 ± 0.6 | 2.2 ± 0.5 | 2.2 ± 0.5 | 0.993 |
| Planning | 2.1 ± 0.7 | 2.2 ± 0.6 | 2.2 ± 0.5 | 0.524 |
| Positive reframing | 1.7 ± 0.7 | 1.8 ± 0.7 | 1.8 ± 0.6 | 0.440 |
| Acceptance | 1.9 ± 0.6 | 1.9 ± 0.5 | 1.9 ± 0.5 | 0.723 |
| Humor | 0.9 ± 0.6 | 1.0 ± 0.7 | 0.7 ± 0.6 | 0.026[1] |
| Religion | 1.3 ± 1.0 | 1.5 ± 1.0 | 1.5 ± 0.5 | 0.598 |
| Emotional support | 2.1 ± 0.6 | 2.1 ± 0.6 | 2.0 ± 0.9 | 0.071 |
| Instrumental support | 2.0 ± 0.8 | 1.9 ± 0.7 | 1.7 ± 0.5 | 0.005[2] |
| Self-distraction | 1.7 ± 0.5 | 1.6 ± 0.6 | 1.7 ± 0.6 | 0.833 |
| Denial | 0.8 ± 0.5 | 0.8 ± 0.7 | 0.8 ± 0.6 | 0.678 |
| Venting | 1.5 ± 0.5 | 1.5 ± 0.6 | 1.3 ± 0.6 | 0.035[3] |
| Substance use | 0.4 ± 0.7 | 0.2 ± 0.6 | 0.4 ± 0.6 | 0.148 |
| Behavioral disengagement | 0.9 ± 0.7 | 0.7 ± 0.7 | 0.8 ± 0.6 | 0.333 |
| Self-blame | 1.6 ± 0.8 | 1.3 ± 0.8 | 1.3 ± 0.7 | 0.031[4] |

**Notes.**
The *p*-value shows the significance of particular characteristics in groups of nurses.
In the case of compared groups, the Kruskal–Wallis test was used.
PSS-10, Perceived Stress Scale; GSES, Generalized Self-Efficacy Scale; Mini-COPE, Inventory for Measuring Coping With Stress.
Corrected the *p*-values for the multiple comparisons:
1) The differences refer to groups ≤ 5 and ≥ 20 ($p = 0.068$)
2) The differences refers to groups ≤ 5 and ≥ 20 ($p = 0.014$)
3) The differences refers to groups ≤ 5 and ≥ 20 ($p = 0.0482$)
4) The differences refers to groups ≤ 5 and ≥ 20 ($p = 0.038$).

status ($\beta$ −0.145, $p = 0.037$) (data in Table 7). A higher stress level was correlated with shorter work experience, poorer health self-assessment, and more working hours.

## DISCUSSION

### Stress during the COVID-19 pandemic

During the COVID-19 epidemic, health professionals, directly engaged in fight against COVID-19, experience chronic fatigue and stress (*Hammami et al., 2021*). Medical workers also display compassion fatigue, burnout (*Gawrych, 2021*; *Neto et al., 2020*), and mental exhaustion (*Alharbi, Jackson & Usker, 2020*).

The results of the original research show that a higher stress level was related to nurses' shorter work experience, poorer self-assessment of health status, and a higher number of working hours. *Ali et al. (2020)* also emphasize that younger American practice nurses declare higher stress than older ones. They point out that younger nurses react more

**Table 5 The level of stress, coping strategies and sense of self-efficacy by the respondents' professional role (n = 220).**

| Variables studied | Nursing role | | |
| --- | --- | --- | --- |
| | Practice nurse (n = 177) | Management nurse (n = 43) | p-value |
| PSS-10, mean ± SD (0–40) | 24.7 ± 3.5 | 25.2 ± 3.2 | 0.268 |
| GSES, mean ± SD (10–40) | 28.9 ± 3.5 | 30.0 ± 3.8 | 0.066 |
| Mini-COPE – strategies of coping with stress (0–3): | | | |
| Active coping | 2.2 ± 0.6 | 2.4 ± 0.6 | 0.110 |
| Planning | 2.2 ± 0.6 | 2.4 ± 0.5 | 0.018 |
| Positive reframing | 1.8 ± 0.7 | 1.8 ± 0.7 | 0.073 |
| Acceptance | 1.6 ± 0.5 | 1.6 ±0.6 | 0.397 |
| Humor | 0.9 ± 0.6 | 0.7 ± 0.6 | 0.124 |
| Religion | 1.5 ± 1.0 | 1.5 ± 0.9 | 0.756 |
| Emotional support | 2.1 ± 0.6 | 2.1 ± 0.6 | 0.537 |
| Instrumental support | 1.9 ±0.7 | 1.7. ±0.7 | 0.062 |
| Self-distraction | 1.7 ± 0.6 | 1.6 ± 0.6 | 0.394 |
| Denial | 0.8 ± 0.6 | 0.9 ± 0.7 | 0.778 |
| Venting | 1.5 ± 0.6 | 1.3 ± 0.5 | 0.033 |
| Substance use | 0.4 ± 0.6 | 0.3 ± 0.6 | 0.604 |
| Behavioral disengagement | 0.9 ± 0.7 | 0.6 ± 0.6 | 0.032 |
| Self-blame | 1.4 ± 0.8 | 1.2 ± 0.7 | 0.118 |

**Notes.**

The p-value shows the significance of particular characteristics in groups of nurses.

In the case of compared groups, the Mann–Whitney U-test was used

PSS-10, Perceived Stress Scale; GSES, Generalized Self-Efficacy Scale; Mini-COPE, Inventory for Measuring Coping With Stress.

emotionally, which results from their lack of experience, resources, or supervision. Nurses also report the lack of psychological support.

The analysis of original research results showed that the place of providing nursing care, connected with the direct risk of SARS-CoV-2 infection, was not related to the level of perceived stress. Chinese researchers also reported they did not find a difference in the prevalence of anxiety and depression among the nurses they studied depending on whether they had direct contact with patients (and hence, were at risk of infection) or not. The authors suggest that nurses who do not work directly at COVID units need attention and support too (*Huan Xiong, Yi & Lin, 2020*). Similarly, a study carried out among Indonesian nurses working at different health care institutions during the COVID-19 pandemic (1/3 only involving direct contact with patients infected with SARS-CoV-2) showed nurses' high occupational burden and a high risk of stress.

Interesting in this context is a study of Canadian nurses, which shows that fears connected with insufficient supply of and access to PPE, personnel shortages, long working hours, care for patients, risk of infection, and isolation from family and friends are a source of stress for nurses (*Doolittle, Anderssen & Perreaux, 2020*). In addition, nurses sometimes experience discrimination and negative treatment by the society due to the fear that they may spread the virus (*Basit & Peni, 2021*; *Stelnicki, Carleton & Reichert, 2020*; *Loeb et al.,*

**Table 6 The level of stress, coping strategies and sense of self-efficacy by the respondents' professional role (n = 220).**

| Variables studied | Working area | | |
| --- | --- | --- | --- |
| | Primary care health (**n = 83**) | Clinical care (**n = 137**) | **p-value** |
| PSS-10, mean ± SD (0–40) | 24.7 ± 3.2 | 24.8 ± 3.6 | 0.771 |
| GSES, mean ± SD (10–40) | 28.6 ± 3.4 | 29.4 ± 3.7 | 0.044 |
| Mini-COPE – strategies of coping with stress (0–3): | | | |
| Active coping | 2.1 ±0.5 | 2.3 ±0.6 | 0.068 |
| Planning | 2.1 ± 0.5 | 2.2 ± 0.6 | 0.024 |
| Positive reframing | 1.8 ± 0.7 | 1.8 ± 0.7 | 0.438 |
| Acceptance | 1.6 ± 0.5 | 1.6 ± 0.6 | 0.934 |
| Humor | 0.8 ± 0.6 | 0.9 ± 0.6 | 0.509 |
| Religion | 1.4 ± 1.0 | 1.5 ± 1.0 | 0.710 |
| Emotional support | 2.0 ± 0.6 | 2.1 ± 0.6 | 0.123 |
| Instrumental support | 1.7 ± 0.7 | 1.9 ± 0.7 | 0.088 |
| Self-distraction | 1.7 ± 0.6 | 1.7 ± 0.6 | 0.906 |
| Denial | 0.9 ± 0.7 | 0.8 ± 0.7 | 0.658 |
| Venting | 1.4 ± 0.6 | 1.9 ± 0.6 | 0.374 |
| Substance use | 0.4 ± 0.6 | 0.3 ± 0.6 | 0.390 |
| Behavioral disengagement | 0.9 ± 0.6 | 0.8 ± 0.7 | 0.041 |
| Self-blame | 1.4 ± 0.7 | 1.4 ± 0.8 | 0.566 |

Notes.
The p-value shows the significance of particular characteristics in groups of nurses.
In the case of compared groups, the Mann–Whitney U-test was used
PSS-10, Perceived Stress Scale; GSES, Generalized Self-Efficacy Scale; Mini-COPE, Inventory for Measuring Coping With Stress.

*2004*; *Shanafelt, Ripp & Trockel, 2020*). Other authors also observe that most nurses express some anxiety about the health of their family members and friends connected with the risk of infecting them (*Ali et al., 2020*), which may reduce their social contacts, causing a gap in the received social support, whereas the levels of anxiety, stress, and sense of self-efficacy are largely dependent on the received social support (*Cai et al., 2020*).

## Sense of self-efficacy in the face of challenges of the COVID-19 pandemic

Another area investigated in this work was the sense of self-efficacy, which determines many aspects of human behavior (*Bandura, 2004*) and is associated with the sense of control of one's own actions (*Juczyński, 2012*). A study carried out during the COVID-19 pandemic in Jordan showed that the participating nurses had a moderate level of sense of self-efficacy and self-confidence in interactions with patients. The researchers found positive correlations between the sense of self-efficacy, self-confidence, and nurse–patient interactions depending on work experience, academic qualifications, and professional roles (*Abu Sharour et al., 2021*). Other authors report that the sense of self-efficacy is correlated with mental health, resilience, and job burnout (*Hsieh, Wang & Ma, 2019*; *Yu et al., 2019*). It is also an important factor of predicting a nurse's willingness to care for patients with infectious diseases (*Valentina Simonetti et al., 2021*).

**Table 7** The level of stress, coping strategies and sense of self-efficacy by the respondents' professional role ($n = 220$).

| PREDICTOR | β | p | 95% Cl | 95% Cl |
|---|---|---|---|---|
| Age | 0.157 | 0.118 | −0.040 | 0.356 |
| Work experience | −0.250 | 0.014 | −0.450 | −0.050 |
| Education | −0.020 | 0.760 | −0.039 | 0.028 |
| Average number of working hours/month | 0.156 | 0.015 | 0.029 | 0.283 |
| Assessment of the atmosphere at work | −0.125 | 0.077 | −0.282 | 0.014 |
| Assessment of security at work | −0.048 | 0.506 | −0.190 | 0.094 |
| Health self-assessment | −0.145 | 0.037 | −0.310 | −0.009 |
| Working area Primary health care nurse Clinical care nurse | −0.072 | 0.267 | −0.208 | 0.058 |
| Nursing role Practice nurse Management nurse | 0.115 | 0.104 | −0.030 | 0.322 |
| Does the employer provide information on the current guidelines/recommendations concerning SARS-CoV-2? | −0.032 | 0.650 | −0.194 | 0.121 |
| GSES scale | 0.007 | 0.914 | −0.124 | 0.138 |

**Notes.**
$R^2 = 0.161$; $F = 3.32$; β - standardized regression coefficient.

The results of the original research show that nurses had an average level of sense of self-efficacy, although the ones who worked at hospitals had higher self-confidence in coping with challenges than did the ones from outpatient facilities. Presumably, working at hospital involves teamwork, which ensures better information flow and an opportunity of training and support, which may be connected with more effective coping in difficult situations. Research carried out at different Italian hospital units during the COVID-19 pandemic showed that women are more susceptible to a low sense of self-efficacy than men ($p < 0.05$) (and in the original study, 96.4% were women). The authors believe that the diagnosed mental problems may lead to the PTSD in the future (*Valentina Simonetti et al., 2021*). Previous research results also show significant correlations between the sense of self-efficacy and mental health (*Hu et al., 2020*) and well-being (*Yuksel, Bayrakci & Bahadır-Yilmaz, 2019*).

## Strategies of coping with stress connected with the COVID-19 pandemic

Coping strategies are important for nurses to handle the COVID-19 pandemic (*Bartzik, Aust & Corinna Peifer, 2021*). Despite the critical situation, analyses carried out by many researchers all over the world during the COVID-19 pandemic show that nurses have enough resources to employ constructive coping strategies (*Sehularo et al., 2021*). For example, a study carried out in China at the beginning of the pandemic showed that nurses displayed quite intensive reactions to the crisis they were experiencing, concentrating on problem-focused strategies rather than on emotions (*Huang, Xu & Liu, 2020*; *Xu & Zhang, 2020*). It was also pointed out that the more tension the participants felt, the more often they

chose emotion-oriented strategies, whereas fear motivated them to display problem-solving attitudes (*Śniegocka & Śniegocki, 2014*; *Siemianowska, Podsiadły & Ślusarz, 2018*).

A study carried out before the pandemic among Polish nurses with the use of a multidimensional inventory to measure coping with stress (COPE) showed that active coping and support seeking strategies weredominant. Another study carried out before the pandemic showed that Polish nurses usually chose active coping, planning, self-distraction, seeking emotional support, positive reframing, and development (*Śniegocka & Śniegocki, 2014*; *Siemianowska, Podsiadły & Ślusarz, 2018*).

The analysis of the results of this study, carried out during the pandemic, shows that younger nurses tended to choose instrumental support (which is an adaptation strategy), but also sense of humour, self-blame and venting (strategies referring to the sense of psychological discomfort). According to *Juczyński & Ogińska-Bulik (2012a)*, humor and venting are less effective in coping with stress, although they are very useful in some situations. On the other hand, *Bartzik, Aust & Corinna Peifer (2021)* believe that the sense of humor and appreciation are two resources that help nurses deal with the COVID-19 pandemic. Literature data also show that the most common way of reducing negative consequences of distress and maintaining the sense of security is access to reliable information (*Heitzman, 2020*; *Folkman & Moskowitz, 2004*; *Lai et al., 2020*; *Chen et al., 2020*). It is worth stressing that most of the studied nurses declared they had constant access to the current information they needed to provide care.

With respect to the workplace, it was shown that practice nurses working directly with patients, regardless of the risk of contact with SARS-CoV-2, tended to choose the strategies of behavioral disengagement and venting, which proves their sense of mental discomfort, while nurse managers significantly more often chose the strategy of planning (a problem-oriented strategy). In addition, clinical nurses were more often observed to choose the strategy of planning, which is a problem-oriented, adaptation strategy arising from the need to reduce the perceived stress (*Juczyński & Ogińska-Bulik, 2012a*), as compared to nurses working at outpatient health care facilities, who significantly more often chose the strategy of behavioral disengagement. A study on Croatian nurses also shows that in the time of the COVID-19 pandemic, nurses use the avoidance and positive reappraisal coping style much more often than do physicians. Furthermore, with respect to age groups, that study shows that individuals under 40 use avoidance coping techniques more often (*Salopek-Žiha, et al., 2020*).

It seems that in the search for possible solutions to the psychological problems occurring in nurses during the pandemic, an important role may be played by the management staff, who should organize regular meetings to reflect on the problems faced by the nurses (*Oktovin, Basit & Peni, 2021*). This is confirmed by a qualitative descriptive study carried out at a hospital in Wuhan, China (the epicenter of the COVID-19 pandemic), concerning the psychological changes occurring in front-line nurses. The researchers highlighted an important role of nursing leaders in supporting nurses and providing psychological help in adaptation to changes (*Zhang, Huang & Wei, 2020*). Similarly, *Shanafelt, Ripp & Trockel (2020)* emphasize that health care workers would like to receive support from their employers through open communication, transparency, comprehensive evaluation of

organizational risk, sufficient and appropriate staffing, access to PPE, trainings, and ongoing psychological assistance. Indonesian researchers also emphasize the role of management in improving personnel's working conditions through support, analyzing workload, ensuring adequate staffing, organizing meetings to solve ongoing problems, and ensuring psychological consultations as needed (*Basit & Peni, 2021*; *Chirico, Nucera & Magnavita, 2021*; *Oktovin, Basit & Peni, 2021*). According to authors from Afghanistan, facilitating continuous and comprehensive support mechanisms aimed at protecting nurses' mental health is of great importance during pandemics (*Nadeem et al., 2021*). The need to educate medical staff in ways of coping with stress is even more obvious now, since the pandemic has shown a deficit in this regard (*Ali et al., 2020*; *Salopek-Žiha, et al., 2020*, *El-Hage et al., 2020*).

### Study limitations

The study definitely has some limitations. First of all, it was a cross-sectional study based exclusively on self-assessment. Although the standardized research tools adapted into Polish that were used in this study are sensitive instruments designed to detect various behaviors and attitudes, all of them concentrate on the respondents' subjective feelings, not on objective criteria, which creates the risk of false positive results. Second, the studied group came from one selected macro region, which makes it impossible to generalize the results to the entire Polish population. Due to the epidemic threat, the study was conducted on-line, which may have affected the response rate (55%). Despite these limitations, the results of the study may be the starting point for further research on the influence of the difficult situation of epidemic threat on nurses' professional functioning. This also shows the potential goals and tasks for the management staff connected with the development of appropriate relationships and attitudes as well as ensuring the personnel safe working conditions during the pandemic.

## CONCLUSION

The higher level of stress of the participating nurses during the COVID-19 pandemic was significantly correlated with shorter work experience, more hours worked a month, and lower health self-assessment. The kind of coping strategies chosen by the respondents was related to their age and role served in the health care system. Younger nurses significantly more often chose coping strategies such as instrumental support, sense of humor, venting, and self-blame, while the management staff more often chose planning. Nurses working at hospitals displayed a higher level of effective coping.

The results of the research show that special attention must be given to younger nurses and those with shorter work experience. It is especially important to monitor nurses' working time and health status. The nurses working at outpatient health care institutions need more attention, support, and information, and managers may play a significant role in improving the mental health of the nursing staff.

### Funding
This work was supported by the Medical University of Bialystok in Poland (SUB/3/DN/21/003/3310). The funders had no role in study design, data collection and analysis, decision to publish, or preparation of the manuscript.

### Grant Disclosures
The following grant information was disclosed by the authors:
The Medical University of Bialystok in Poland: SUB/3/DN/21/003/3310.

### Competing Interests
The authors declare there are no competing interests.

### Author Contributions
- Matylda Sierakowska conceived and designed the experiments, performed the experiments, analyzed the data, authored or reviewed drafts of the paper, and approved the final draft.
- Halina Doroszkiewicz conceived and designed the experiments, performed the experiments, analyzed the data, prepared figures and/or tables, authored or reviewed drafts of the paper, and approved the final draft.

### Data Availability
The raw measurements are available in the Supplementary File.

### Supplemental Information
Supplemental information for this article can be found online at http://dx.doi.org/10.7717/peerj.13288#supplemental-information.

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
