# Peer review of "Stress coping strategies used by nurses during the COVID-19 pandemic"

_PeerJ, doi:10.7717/peerj.13288_

## Round 0.1 · original submission · Minor Revisions

Thank you for submitting this manuscript. Please pay careful attention to the suggestions made by reviewers 1,3 and 4. Specifically, please make sure your research questions(s), aims and hypotheses are clearly articulated and then use this as a guide for your findings and discussion section. Also, I believe the manuscript will be stronger if you are able to even more strongly articulate why the stressors outlined here in this research are different from stressors generally found in the nursing profession.

Finally, based on all of the reviewers' suggestions, your manuscript will likely benefit from an English language copy editor.

·

Basic reporting

1. The tables are professionally prepared, however the average age of respondents should be corrected (it is 3.3, and should be, as in the description 43.3).

Experimental design

2. The authors provide a well-formulated research goal in line with the conducted analyzes, nevertheless they should complete the research problem and the assumed hypothesis.

Validity of the findings

3. The first part of the conclusion is correct, but the part of the conclusion from line 423 to line 427 has no justification in the analyzed data.

Reviewer 2 ·

Basic reporting

The paper appears to be well-written but a modest improvement in the use of professional English language may be necessary. For example, the first sentence of the Introduction Section states, “The risk of SARS-CoV-2 infection, as well as the sense of helplessness, loss of control and unpredictability, are among many elements that currently affect nurses' professional functioning.” This is a vague and not much informative opening statement and also lacks a proper reference.
A better rendering could be “ The risk of getting SARS-CoV-2 infection, worries about exposing loved ones, anxiety and frustration, emotional and physical exhaustion, burn out, a feeling of being overwhelmed, and struggles and challenges with parenting are a few among many factors that affect nurses’ personal lives and professional functioning.”

Experimental design

The research hypothesis/ research question(s) and objectives are not clearly articulated. The sampling type, sampling method, inclusion/exclusion criteria, and study population are not well defined.

Validity of the findings

The validity of the finding is questionable. There is no baseline study conducted prior to the advent of the pandemic to compare with the results of this study conducted during the pandemic. The study appears to project mere speculation that all the stress faced by the nursing professionals is attributable to the pandemic. Nursing professionals are known to work under high stress even absent the added burden of the pandemic and this study does not quantify how much of the stress is caused by having to work during this pandemic. Moreover, the study does not provide the variance of the outcome variable expressed by each independent variable. Despite the title of the study, “Stress coping strategies used by nurses in the time of the COVID-19 pandemic,” the authors do not provide as part of the results of their study, what established stress coping strategies are employed by the sample population (There is no mention of the methods of coping used by these nurses.)

Additional comments

The conclusions of the study appear to be causally unrelated to the stress of having to work during a pandemic. Based on many years of lived professional experience, this reviewer believes that the conclusions of this study would be valid in the absence of the COVID-19 epidemic, given the already high-stress levels among the nursing care providers.

Reviewer 3 ·

Basic reporting

1. In the present study, stress coping strategies used by nurses during the COVID-19 pandemic are described.
2. Throughout the paper, clear, unambiguous, and professional language is used, with only minor grammatical errors.
3. In this paper, there is sufficient background information to show how the work fits into the broader field of knowledge.
4. In general, the paper is self-contained and includes all results relevant to the hypothesis.

Experimental design

Experimental design:

Responses were collected from 220 nurses providing health care in both inpatient and outpatient settings in Poland's north-eastern region. The study was conducted online using the diagnostic poll method and a survey questionnaire containing the General Self-Efficacy Scale (GSES 10-40), the MINI-COPE questionnaire (0-3), and the Perceived Stress Scale (PSS-10).

The paper reports the following major findings:
- Major findings related to stress are:

1. There was a high level of stress among the nurses (PSS-10 - 20.9),

2. The stress was related to:
2.1. Work experience as a nurse (β -0.250, p=0.014),
2.2. Hours worked per month (β 0.156, p=0.015), and
2.3. Self-assessment of health (β -0.145, p=0.037))
3. There was an average sense of self-efficacy (GSES - 29.1), and it depended on the workplace (p=0.044).

- Strategies for coping with stress (Mini-COPE):
1. The younger nurses mentioned
1.1. Venting (p=0.010),
1.2. Instrumental support (p=0.011),
1.3. Sense of humor (p=0.013) and
1.4. Self-blame (0.031).

Strengths of the paper:

1. The paper includes clearly describes the introduction, methods, and major findings.
2. The paper performs sound statistical analysis and reports the details enough to replicate the study.
3. Conclusions are supported by the evidence presented in the paper.
4. There is a knowledge gap about the factors that induce stress and the coping mechanisms used by nursing staff during the COVID-19 pandemic. The article focuses specifically on this aspect.

For suggestions regarding improvement please see the comments below.

Validity of the findings

Suggestion for improvement:
1. The paper contains multiple typographical and grammatical errors. Proofreading can easily correct minor grammatical errors.
2. The study was conducted online, so a 55% response rate may have introduced some bias to the results. Stressed out respondents may have participated more in the survey, or vice versa.
3. As the paper compares multiple tests and results. However, these multiple statistical tests were not corrected for the multiple comparisons. I would suggest correcting the p-values for the multiple comparisons to prevent false-positive results. Although it won't affect negative findings, it will rule out false-positive findings. Benjamini and Hochberg's false discovery rate (FDR) correction method will be useful here for correcting the FDR. In order to establish the reliability of the results, it is vital to correct the p-values.

Additional comments

Overall comment: The paper is generally well written and has a logical flow. It provides enough details for reproducibility and many of its methods are sound.

·

Basic reporting

Basic Reporting
• Thank you for this relevant journal on an important topic. I would recommend updating the title to align with English language flow. The title, as it currently reads, is slightly awkward. For example, it would read better as “Stress coping strategies used by nurses during the COVID-19 pandemic”. I suggest you have a colleague who is proficient in English and familiar with the subject matter review your manuscript, or contact a professional editing service for consistency in the English language. Some other notable text needing revision are mentioned below:
o In line 4, the running title, please incorporate the concept of stress or coping strategies as this is the main concept of the article.
o In line 37, a stand alone paragraph of one sentence is not appropriate.
o Line 185 is an incomplete sentence.
o In the supplemental file of informed consent, the Polish to English translation is poorly completed.
• In the abstract, please follow the journal guidelines (Methodology/Methods is not bolded). Also:
o Subheadings must be bold, followed by a period, and start a new paragraph e.g.
Background. The background section text goes here...
• The Raw data is a mixture of Polish and English and with 220 responses, it is difficult to remember which translation applies to the English translation. As the article is submitted in the English language, the raw data should also be provided 100% in English.
• An extensive description of the PSS-10, Mini-COPE and GSES are not needed, in my opinion, as these are well-known and published questionnaires. A simple description and reference would be perfectly acceptable in this instance in the setting of such well-known tools.
• In the Discussion, please avoid 1 or 2 sentence paragraphs. Appropriate paragraph length is 3-5 sentences.
• The literature search is somewhat outdated. Few articles are included from 2021 and there are many new relevant articles that have been published in 2021 that can strengthen the literature review and discussion even further.

Experimental design

I agree that the research question is well defined, meaningful and relevant and fills a needed gap in knowledge. The methods and data analysis and appropriate ethics approvals are well-described.

Validity of the findings

In Table 1, if all 220 answered the question on M or F, the percentage should be 96.4, not 96.6. Kindly confirm all respondents answered this questions. 212/220 = 96.4%
In Table 2, please confirm this is correct
<!--[if !msEquation]--> <!--[endif]--> x¯ - mean, ± SD – standard deviation. Is this a typo?

Additional comments

I commend the authors for their work in this field in offering a comprehensive review of Polish nurses, a notable group not previously reported extensively in the literature. The statistical analysis is strong and well-reported and explained. For the main area of improvement, it is in the flow of English I have noted above) which should be improved upon before Acceptance as well as updating the literature review including more articles from 2021.
Finally, please provide all raw data fields in English.

---

## Round 0.2 · accepted · Accept

Thank you for addressing the reviewers questions, suggestions and concerns. We appreciate your patience in this iterative process, congratulations.

·

Basic reporting

No comment

Experimental design

No comment

Validity of the findings

No comment

Additional comments

No comment

Reviewer 3 ·

Basic reporting

NA

Experimental design

NA

Validity of the findings

NA

Additional comments

My points raised during the previous review have been addressed by the authors. I have no further suggestions.

·

Basic reporting

The authors have made all suggested changes. The article reads very well. No additional changes requested.

Experimental design

The experimental design is appropriate and research question well defined. No changes requested.

Validity of the findings

All data has been provided and conclusions well-stated. No additional changes are requested.

Additional comments

Thank you to the authors for their very thoughtful and thorough revision to the manuscript. I fully support moving forward to publication of this important paper.